# Metadherin Regulates Inflammatory Breast Cancer Invasion and Metastasis

**DOI:** 10.3390/ijms24054694

**Published:** 2023-02-28

**Authors:** Gabriela Ortiz-Soto, Natalia S. Babilonia-Díaz, Mercedes Y. Lacourt-Ventura, Delmarie M. Rivera-Rodríguez, Jailenne I. Quiñones-Rodríguez, Mónica Colón-Vargas, Israel Almodóvar-Rivera, Luis E. Ferrer-Torres, Ivette J. Suárez-Arroyo, Michelle M. Martínez-Montemayor

**Affiliations:** 1Department of Biochemistry, Universidad Central del Caribe-School of Medicine, Bayamón, PR 00960, USA; 2Department of Clinical Anatomy, College of Osteopathic Medicine, Sam Houston State University, Conroe, TX 77304, USA; 3Department of Anatomy and Cell Biology, School of Medicine, Universidad Central del Caribe, Bayamón, PR 00960, USA; 4Department of Mathematical Sciences, University of Puerto Rico at Mayagüez, Mayagüez, PR 00681, USA; 5Department of Pathology and Laboratory Medicine, Hospital Interamericano de Medicina Avanzada (H.I.M.A.)—San Pablo Caguas, Caguas, PR 00725, USA; 6Department of Immunopathology, Hato Rey Pathology Associates Inc. (HRPLABS), San Juan, PR 00936, USA

**Keywords:** metadherin, inflammatory breast cancer, invasion, metastasis

## Abstract

Inflammatory breast cancer (IBC) is one of the most lethal subtypes of breast cancer (BC), accounting for approximately 1–5% of all cases of BC. Challenges in IBC include accurate and early diagnosis and the development of effective targeted therapies. Our previous studies identified the overexpression of metadherin (MTDH) in the plasma membrane of IBC cells, further confirmed in patient tissues. MTDH has been found to play a role in signaling pathways related to cancer. However, its mechanism of action in the progression of IBC remains unknown. To evaluate the function of MTDH, SUM-149 and SUM-190 IBC cells were edited with CRISPR/Cas9 vectors for in vitro characterization studies and used in mouse IBC xenografts. Our results demonstrate that the absence of MTDH significantly reduces IBC cell migration, proliferation, tumor spheroid formation, and the expression of NF-κB and STAT3 signaling molecules, which are crucial oncogenic pathways in IBC. Furthermore, IBC xenografts showed significant differences in tumor growth patterns, and lung tissue revealed epithelial-like cells in 43% of wild-type (WT) compared to 29% of CRISPR xenografts. Our study emphasizes the role of MTDH as a potential therapeutic target for the progression of IBC.

## 1. Introduction

Inflammatory breast cancer (IBC) is a very aggressive and rare subtype of breast cancer (BC) with a poor prognosis and accounts for 1–5% of all cases of BC. The 5-year survival rate for people with IBC is 41% compared to non-IBC (nIBC) [1,2,3]. However, a recent study in Puerto Rico reports that the five-year survival rate for IBC patients is 0%, while the three-year survival is 39%, highlighting the urgency of IBC studies [4]. The lethality of IBC is due to the efficient ability of cancer cells to invade the vascular and lymphatic systems and often the absence of a distinct tumor mass [1]. The uncommon nature of this BC subtype, the rare clinical presentation, and the overlap of these atypical signs and symptoms with other diseases complicate its diagnosis and treatment [5]. Although the introduction of multimodal therapy has improved overall survival, it remains poor compared to nIBC [6]. Therefore, studies directed toward finding targets and understanding IBC biology are essential to improve therapeutic strategies and overall patient outcome.

Previously, our group characterized the cell surface proteome of IBC. We identified metadherin (MTDH) overexpression in IBC cell lines vs. non-cancerous mammary epithelial cells, and nIBC cell lines [7]. Subsequently, the presence of MTDH was validated in IBC tissues and within tumor emboli [7], a hallmark of IBC metastasis [1]. MTDH is a single-pass transmembrane protein expressed mainly in the endoplasmic reticulum and the perinuclear space and has been found to colocalize with tight junctions [8,9,10,11,12,13]. MTDH is a cell adhesion molecule and functions as a scaffold protein that interacts with multiple cell networks [13,14,15,16]. The localization of MTDH varies depending on the state of the cell; it is found in the nucleus in non-malignant tissue and in the cytoplasm and cell membrane in malignant tissue, where it interacts with various oncogenic pathways. Some of the pathways associated with MTDH interactions are NF-κB, PI3K/AKT, MAPK, and Wnt/β-catenin [17,18,19,20,21,22,23]. Dysregulation of these pathways involves cell regulatory mechanisms, such as proliferation, migration, and cell survival; which promotes chemoresistance, invasion, and metastasis [8].

Metastasis is the leading cause of death in BC patients [24]. IBC patients have a higher risk of distant metastasis and locoregional recurrence than nIBC patients [25]. More than 30% of IBC patients have metastases at the time of diagnosis [26,27]. Understanding the metastatic processes that influence the aggressiveness and lethality of IBC is crucial to support the multidisciplinary discussion of potential approaches to diagnose and treat IBC. Furthermore, various groups confirm that MTDH plays an important role in BC metastasis [15,28]. The pro-metastasis function of MTDH is due to the interaction of a lung-homing domain of MTDH with an unknown receptor in endothelial cells [15,29] and has been shown to increase expression of adhesion molecules by activating NF-κB [21]. However, the role of MTDH in the invasion and metastasis of IBC remains unknown.

This study investigates the role of MTDH in IBC by assessing its function in cell proliferation, migration, tumor spheroid formation, signaling of oncogenic pathways, tumor progression, and metastasis. We develop isogenic models of the SUM-149 and SUM-190 IBC cell lines using MTDH-CRISPR/Cas9 base knockout for in vitro and in vivo studies. To validate the oncogenic role of MTDH, we also overexpressed MTDH in the non-cancerous mammary epithelial cell line MCF-10A. Our findings show that the absence of MTDH significantly decreases IBC cell proliferation, migration, invasion, and tumor spheroid formation and decreases the expression of various signaling molecules. In addition, we demonstrate that MTDH affects invasion and metastasis. To our knowledge, this study is the first to assess the role of MTDH in the invasion and metastasis of IBC. Thus, this study provides significant new evidence for the field of IBC, contributing to studying possible new targets involved in IBC progression.

## 2. Results

### 2.1. MTDH Is Overexpressed in IBC Cell Line Models Compared to the Non-Cancerous Mammary Epithelial Cell Line, MCF-10A

In a previous study, we demonstrated that MTDH is overexpressed in IBC cells and patient tumor tissues compared to normal tissues and in the tumor emboli. Furthermore, we reported that MTDH overexpression was found in the plasma membrane of IBC cells, specifically in SUM-149 cells, compared to the non-cancerous mammary epithelial cell line, MCF-10A [7]. Moreover, IHC data showed strong cytoplasmic expression in IBC tissues compared to normal tissues [7]. To validate that MTDH is not solely overexpressed in the plasma membrane of IBC cells, whole cell lysates of SUM-149 (triple-negative (TN) IBC) and SUM-190 (HER2+ IBC) cells were immunoblotted. As shown in Figure 1A,B, MTDH is significantly overexpressed in SUM-149 with a 2.6 fold-change (f.ch.) and 3.0 f.ch. for SUM-190 IBC cell lines compared to MCF-10A cells. These findings confirm our results and correlate with previous findings from Zhang et al., where high ratios of HER2 transcripts increased proteomic levels of MTDH [30]. 

### 2.2. MTDH Knockout Results in Reduced IBC Cell Proliferation and Colony Formation

Several publications have highlighted MTDH as an important target in the treatment of BC, including its role in cancer onset and progression [31,32]. Therefore, our objective is to elucidate the cellular and molecular functions that MTDH plays in IBC. We hypothesized that knocking out MTDH in IBC cells would decrease proliferation and their capacity to form colonies. Here, we developed a MTDH CRISPR/Cas9-based knockout cell line (sg-MTDH) using SUM-149 and SUM-190 IBC cells. To test our hypothesis, we compared their expression with wild-type (WT) cells. The knockout of the MTDH gene in SUM-149 and SUM-190 was validated by qPCR and immunoblotting (Figure 2A,B,E,F). IBC sg-MTDH SUM-149 cells show a significant decrease in gene expression levels (−3.44 f.ch.) and protein expression levels (59%) compared to WT. Similarly, sg-MTDH SUM-190 IBC cells showed a significant reduction in gene expression (−3.56 f.ch.) and protein expression levels (67%) compared to WT. To test differences in cell proliferation capacity, we monitored SUM-149 and SUM-190 (WT and sg-MTDH) for 24, 48, and 72 h. The results showed that the proliferative capacity of SUM-149 (26%) and SUM-190 (51%) was significantly reduced at 72 h in the sg-MTDH cell model (Figure 2C,G). Consistent with the proliferation results, the clonogenicity of MTDH-depleted cells was significantly reduced compared to WT cells (Figure 2D,H). Furthermore, we hypothesized that the overexpression of MTDH in normal cells would promote malignant alterations. To investigate the malignant effects of MTDH in normal cells, we overexpressed MTDH in the non-cancerous mammary epithelial cell line, MCF-10A. We confirm the overexpression of MTDH in MCF-10A cells by performing qPCR and immunoblotting for both control cells (Ctrl) and overexpressed cells (OE-MTDH) (Figure 2I,J). The results showed a significant increase in gene expression (8.33 f.ch.) and protein expression (96%) in OE-MTDH cells compared to Ctrl. As predicted, the proliferative capacity of OE-MTDH increased significantly (72%) compared to Ctrl cells. These results suggest that the depletion of MTDH in IBC cells is directly related to the proliferative capacity of IBC cells in short and long periods of time. 

### 2.3. MTDH Knockout Reduces the Size of IBC-Derived Tumor Spheroids

Previous studies have shown that IBC cells can spontaneously form 3D tumorspheres in vitro [33,34,35]. To study whether the decrease in cell proliferation in SUM-149 and SUM-190 was translated into a 3D culture, we tested the ability of cells to form tumor spheroids after 96 h in culture. Representative images of the tumor spheroids of SUM-149 and SUM-190, WT, and sg-MTDH are shown in Figure 3A,D. We observed that the SUM-149 and SUM-190 cells of the sg-MTDH cell model significantly formed the smaller tumor spheroids (Figure 3B,E) derived from IBC cells. However, there were no significant differences in the number of spheroids (Figure 3C,F). These findings suggest that silencing MTDH affects the capacity of WT IBC cells to form larger tumor spheroids, suggesting a potential role of MTDH in the maintenance of tumor spheroid integrity in IBC cells.

### 2.4. MTDH Modulates the Migration and Invasive Capacity of IBC Cells

MTDH overexpression has been associated with increased cancer cell metastasis [10]. Therefore, we evaluated the effects of MTDH knockout on the migration and invasion potential of IBC cells. We performed a 24 h wound healing assay using SUM-149 and SUM-190 cell lines (WT and sg-MTDH). The results showed a significant reduction in the percent of wound closure of the sg-MTDH cell model in both SUM-149 (79% reduction) and SUM-190 (83% reduction) cells (Figure 4A–D), suggesting that there is a reduction in the IBC cell migration capacity. We also investigated whether MTDH overexpression promotes the migration of non-cancerous cells MCF-10A. The results showed a significant alteration in the ability of OE-MTDH (92% increase) cells to close the wound compared to Ctrl MCF-10A cells (Figure 4E,F). Next, we assessed the effects of the absence of MTDH on IBC cell invasion using a transwell assay with SUM-149 WT and sg-MTDH cells. We observed a significant reduction in sg-MTDH invasion (46%) compared to WT cells (Figure 4G,H). This evidence suggests that MTDH affects the migratory and invasive capacity of IBC cells, confirming a key role in cell motility.

### 2.5. Silencing of MTDH Modulates the Expression of STAT3, NF-κB, and AKT Expression

Since we showed that MTDH depletion plays a key role in proliferation, tumor spheroid formation, and IBC cell motility, we then investigated how key signaling pathways related to these functions are affected. MTDH has been associated with the NF-κB, AKT, and ERK pathways, affecting cancer cell function [8,17,18,36]. Moreover, NF-κB has been found to be hyperactive in IBC when compared to nIBC models [37,38]. Furthermore, because our group and others have demonstrated the importance of the JAK2/STAT3 pathway in the regulation of IBC cellular identity [39], stemness, and tumor progression [40], herein, we also investigated the modulation of STAT3 expression after MTDH knockout. First, we evaluated the expression of the *STAT3* and *NFKB1* genes in both IBC cell models. For SUM-149, there is a trend (*p* = 0.10) of *STAT3* downregulation (−2.00 f.ch.) in sg-MTDH cells. Moreover, the expression of the *NFKB1* gene is significantly reduced by −2.11 f.ch. (Figure 5A) in sg-MTDH cells. In contrast, there were no significant differences in the expression of the *STAT3* or *NFKB1* gene expression in SUM-190 (Figure 5B). 

We also evaluated the protein expression of these signaling molecules by immunoblotting whole cell lysates of SUM-149 and SUM-190 (WT and sg-MTDH) and MCF-10A cells (Ctrl and OE-MTDH) cells (Figure 5C). Densitometry analysis demonstrates that, in SUM-149 sg-MTDH cells, there is a significant downregulation of STAT3, AKT and NF-κB protein expression. No significant differences were observed for JAK2 and ERK (Figure 5D). Similarly, there was a significant reduction in AKT and NF-κB in SUM-190 sg-MTDH cells. However, we also saw a significant reduction in ERK expression. Interestingly, there were no significant differences in STAT3 expression (Figure 5E), which could be related to HER2+ overexpression and changes in the downstream signaling in SUM-190 IBC cells. These findings suggest a possible crosstalk in the regulation of STAT3 and NF-κB total protein expression through MTDH in TN SUM-149 sg-MTDH cells but not in HER2+ SUM-190 sg-MTDH cells. Furthermore, in MCF-10A OE-MTDH cells, there was an upregulation in the protein expression of STAT3 and NF-κB signaling molecules (Figure 5F). These findings validate the possible interaction of a regulation between STAT3 and NF-κB in SUM-149 IBC cells. A crosstalk between STAT3 and NF-κB signaling pathways has previously been described where both proteins regulate distinct and overlapping groups of genes during tumorigenesis [41]. We also verified whether MTDH knockout or overexpression affected the phosphorylation of probed proteins. However, there were no significant differences in the activation of any of the signaling molecules evaluated (Figure 5G–I).

### 2.6. MTDH Editing in IBC Xenograft Models Delays Tumor Development and Decreases STAT3 Expression and Metastasis

MTDH has been correlated with BC progression and poor overall survival in patients [42], but its role in regulating tumor cell proliferation remains controversial [14]. Therefore, we investigated the role of MTDH in the regulation of SUM-149 WT and sg-MTDH tumor growth. We hypothesized that a reduction in MTDH would affect tumor development and progression. 

First, the results showed that the health of the mice was not compromised by the absence of MTDH (Figure 6A). A reduction in tumor size was observed at weeks 1 and 2 in sg-MTDH tumors compared to WT (Figure 6B), and, contrary to what we expected, the absence of MTDH did not reduce tumor volume or tumor weight (Figure 6C). These results demonstrate that the reduction in cell proliferation detected in vitro was not observed in vivo, at least in the primary tumor. Furthermore, H&E staining of lung tissues from WT and sg-MTDH mice revealed the presence of epithelial-like cells. For the WT group, 43% of the mice showed aggregates of epithelial-like cells in the alveolar parenchyma (Figure 6D), as Zhang et al. also described [43]. On the other hand, in the sg-MTDH group, 29% of the mice showed aggregates of epithelial-like cells near the hilum and between the lung arterioles. These results are consistent with the in vitro findings in which MTDH silencing demonstrates a reduction in the invasive capacity of IBC cells. Although no significant changes in primary tumor volume were observed, metastatic lesions were found in a higher percentage of total WT mice than in the sg-MTDH mice group. These findings suggest that what is affected is the invasion and metastatic potential of tumor cells instead of the progression of the primary tumor in vivo. 

To validate that SUM-149 sg-MTDH cells remained with a low expression of MTDH after 10 weeks of study, we performed a qPCR for MTDH gene expression. A significant reduction (−5.52 f.ch.) was observed (Figure 6E) in mouse tumors of the sg-MTDH group compared to WT. We also investigated whether MTDH depletion affected the expression of signaling molecules in sg-MTDH tumors observed in in vitro models. MTDH depletion showed a significant decrease in STAT3 expression, consistent with what we observed in SUM-149 sg-MTDH cells (Figure 6F,G). We also verified whether the reduction of MTDH in sg-MTDH tumors affected the activation of the indicated proteins and there were no significant differences detected (Figure 6H). 

## 3. Discussion

High expression of MTDH has been associated with tumor metastasis, decreased survival outcomes, and higher mortality in female reproductive cancers such as breast, ovarian, and cervical [44]. We previously reported that MTDH is overexpressed in the plasma membrane of IBC cells compared to non-cancerous mammary cells and nIBC cell lines, in addition to its presence in IBC tissues and the tumor emboli [7]. Many studies have shown that lethality of IBC is due to its ability to invade the vascular and lymphatic systems through the tumor emboli (non-adherent clusters of cancer cells), causing the inflammatory phenotype, breast edema, and lymph node metastases [45]. Therefore, studies are needed to identify potential proteins associated with IBC invasion and metastasis. To our knowledge, this is the first study to evaluate the functional role of MTDH in IBC invasion and metastasis. 

Our results demonstrated that MTDH depletion decreased the proliferation of the TN and HER2+ IBC models and their ability to form colonies. At the same time, MTDH overexpression promoted the proliferation of non-cancerous mammary cells. Although MTDH has been correlated with BC proliferation by showing high levels of Ki-67 in tissues, previous studies demonstrated that knocking down of MTDH did not affect the proliferation of MDA-MB-231-LM2 cells, a subcell line of nIBC with a high propensity to lung metastasis [28,46,47], in contrast to our unique findings in IBC cells. We also demonstrate that depletion of MTDH reduced the size of tumor spheroids, suggesting that MTDH could play an important role in their integrity, possibly affecting cell-to-cell interactions. These findings may have a correlation with studies that have demonstrated MTDH as a cell membrane protein important in tight junctions (TJ) during cell–cell adhesion interactions [12]. TJ proteins fundamentally influence cell processes that regulate polarity, differentiation, and migration, all of which are critical steps to cancer progression [48].

Herein, we also established that depletion of MTDH decreased the migration and invasion of IBC cells while overexpression of MTDH promoted the migration of non-cancerous cells. Studies demonstrate that MTDH is relevant in cancer cell invasion, since Matrigel invasion assays have shown that different cancer cells, such as hepatocellular carcinoma and glioma cells, display an increased invasive ability through MTDH overexpression [36]. Additionally, the upregulation of MTDH increased the invasion of cancer cells by upregulation of matrix metalloprotease enzymes (MMPs), specifically MMP-2 and MMP-9 [49,50]. Interestingly, IBC is known to ubiquitously express E-cadherin in primary tumors, tumor emboli, and is associated with metastasis [51,52]. Studies have shown that E-cadherin increases the invasion capacity of the SUM-149 cell line by increasing the levels of MMPs [53]. E-cadherin promotes the dissemination of IBC cells by maintaining embolus integrity through cell-to-cell interactions. Studies suggest that E-cadherin plays a key role in a passive metastatic mechanism by which IBC cells invade the circulatory system in clusters, rather than spreading as single cells [33,54]. Therefore, our findings can have a potential correlation with the cellular and molecular functions of E-cadherin, as the migration and invasion capabilities of MTDH-depleted cells were impaired, as well as the integrity of the spheroids. More studies could be conducted to better understand the relationship between MTDH depletion and E-cadherin function in IBC since MTDH was also found present in tumor emboli in our previous studies [7]. 

In the current study, we also assessed the expression of signaling pathways that play an important role in the regulation and interactions of MTDH, such as AKT, ERK, and NF-κB [36]. We demonstrate that the depletion of MTDH negatively regulates the genetic expression of NF-κB (*NFKB1*) and the expression of NF-κB, AKT, ERK, and STAT3 in a TN IBC model. In a HER2+ IBC model, no changes in gene expression were observed and only a down-regulation of NF-κB, AKT, and ERK proteins was detected. The contrasting results in gene expression between the IBC models could be explained by previous studies which have demonstrated that the amplification or overexpression of HER2 promotes the activation of NF-κB and STAT3 in contrast to TNBCs [55,56]. Interestingly, no changes in the phosphorylated proteins were observed for both isogenic models. Therefore, we propose that these pathways are being affected via post-transcriptional regulation. Previous studies have demonstrated that MTDH interacts with the cyclic AMP-responsive element binding protein (CBP), which is a NF-κB coactivator that serves as a bridge element for NF-κB, CBP, and the basal transcription machinery, enhancing migration and invasion processes [17,57]. MTDH has also been shown to interact with NF-κB by translocating into the nucleus and combining with the p65 subunit of NF-κB and promoting the expression of downstream genes as cell adhesion molecules (i.e., ICAM-2, ICAM-3, selectin P ligand, selectin E, selectin L), toll-like receptor TLR4 and TLR5, FOS, JUN and cytokines IL-8 that are involved in tumor progression and metastasis [17].

Furthermore, we evaluated the STAT3 pathway due to its importance in the regulation of IBC growth and tumor progression described by us [40] and others [39]. In IBC, STAT3 has been implicated to play a crucial role, because IL-6, an inflammatory cytokine that activates the STAT signaling pathway, is up-regulated in IBC tumors compared to nIBC [58]. It is known STAT3 collaborates with NF-κB to promote cancer development and progression [41]. STAT3 and NF-κB, as two important transcription factors, bind cooperatively in a subset of gene promoters to collaboratively induce gene expression during tumorigenesis. More specifically, members of the NF-κB family such as RelA can physically interact with STAT3, and their association can modify their transcriptional activity, promoting tumorigenesis [59]. Currently, there are no studies demonstrating a direct relationship between MTDH and STAT3, but a recent study showed that treating lung cancer cells with a mushroom extract decreased the expression of both MTDH and STAT3, inhibiting proliferation and metastasis [60]. Based on our results, we suggest that MTDH could have a potential effect on STAT3 expression through the regulation of the NF-κB in this TN-IBC model as it has been suggested in other models [61,62]. On the other hand, we did not report any changes in STAT3 expression in the HER2+ IBC model. According to a study by Du et al., up-regulation of MTDH is associated with reduction of the phosphatase and tensin homolog deleted on chromosome 10 (PTEN) and resistance to trastuzumab in HER2+ BC [63]. The same study concludes that MTDH modulates the PTEN-PI3K/AKT pathway through a NF-κB-dependent pathway [63]. These findings can potentially correlate with our results, which suggest that both AKT and NF-κB are affected by MTDH depletion but not STAT3 in HER2+ IBC cells. Additional studies are needed to understand the role of MTDH in NF-κB and STAT3 for IBC models. 

Given that our in vitro results suggested that MTDH knockout would decrease tumor growth, we developed IBC xenograft models using WT and MTDH knockout cells. Our results show that MTDH depletion only reduced tumor volume in the early stages of the study, however the reduction was not sustained. These discrepancies suggest the involvement of interactions between the cancer cells and additional components of the tumor microenvironment in the in vivo model that might compensate the growth response [64]. Nonetheless, our results demonstrated that upon MTDH silencing there was reduced metastasis to lung tissue of mice. Interestingly, studies have described that MTDH contains an extracellular lung-homing domain that mediates BC cells to form lung metastasis [15]. In addition, another study demonstrated that MTDH knockdown reduced the adhesion of nIBC cells to lung microvascular endothelial cells, as well as to the bone marrow [28]. In our study, we show that MTDH knockout decreases lung metastasis of MTDH depleted TN IBC cells (29%) compared to WT (43%). Based on these findings, we propose that a lack of MTDH delays the proliferative capacity of tumor initiating cells, which affects the initial stage of tumor development and does not affect primary tumor growth but regulates invasion and metastasis. Furthermore, we evaluated NF-κB and STAT3 in tumor xenograft models where MTDH remained depleted after 10 weeks. We found that there is a reduction in NF-κB and STAT3 proteins, which was observed in the in vitro model using the same TN IBC cells. 

In this study, we confirm and validate that MTDH promotes metastasis as suggested by other studies [15,28]. Therefore, our results suggest a functional role for MTDH in the aggressiveness of IBC by affecting cell proliferation, tumor spheroid formation, migration, and invasion. Furthermore, we showed for the first time how MTDH depletion affects the STAT3 and NF-κB signaling pathways in IBC models and a potential crosstalk between pathways. We also demonstrated the effects of MTDH depletion on early tumor development and metastasis in xenograft models. In summary, this study shows the potential of MTDH as a possible target to better understand the progression and metastasis of IBC.

## 4. Materials and Methods

### 4.1. Cell Culture and Reagents

The SUM-149 and SUM-190 cell lines were obtained from BioIVT (Westbury, NY, USA) and cultured in Ham’s F-12 Nutrient Medium (Gibco/Life Technologies, Waltham, MA, USA) according to the manufacturer’s instructions. SUM-149 cells were supplemented with 10% fetal bovine serum (FBS) (Corning^®^, Corning, NY, USA) and SUM-190 cells were supplemented with 2% FBS (Corning^®^), 1 g/L bovine serum albumin (BSA) (Rockland Immunochemicals, Pottstown, PA, USA) and insulin-transferrin-sodium (ITS) (Sigma-Aldrich, St. Louis, MO, USA). HEK-293 and MCF-10A cells were obtained from ATCC^®^ (Manassas, VA, USA). The human embryonic kidney cell line HEK-293 (ATCC^®^ CRL-1573^TM^) was cultured in Dulbecco’s Modified Eagle Medium (DMEM) (Gibco/Life Technologies) and supplemented with 10% FBS. The human non-cancerous mammary epithelial cell line MCF-10A (ATCC^®^ CRL-10317^TM^) was cultured in DMEM/Ham’s F-12 (Gibco/Life Technologies) with 10% horse serum (HS) (Sigma-Aldrich) supplemented with 20 ng/mL of epidermal growth factor (EGF), Cholera Toxin B, Hydrocortisone solution, HEPES and insulin (Sigma-Aldrich) as described in [65,66]. Cells were regularly tested to ensure they were free of mycoplasma infection using the Mycoplasma Detection Kit (ASB-1310001, Nordic BioSite AB, Sweden). All cell lines were genotyped for authenticity using the short tandem repeat (STR) profile and interspecies contamination testing services from IDEXX BioResearch (Columbia, MO, USA).

### 4.2. Plasmids, Lentiviral Particle Generation, and Mammalian Cells Transduction

Plasmids: The MTDH-CRISPR/Cas9 vectors were constructed in the Duke Functional Genomics Shared Resource at Duke University (Durham, NC, USA) using the lentiCRISPR/Cas9 lentiCRISPRv.2.0 backbone plasmid 52961 from Addgene (Watertown, MA, USA). The MTDH overexpression vector (LV-h-MTDH ORF-GFP) and the empty vector (LV-GFP) were constructed and purchased from VectorBuilder (Chicago, IL, USA). Vectors for packaging (PCMV delta R8.2) and envelopes (pMD2.G) of viral particles were purchased from AddGene.

Lentiviral Particles Generation: HEK-293 (2 × 10^6^) cells were seeded in 60 mm culture plates and transiently transfected with the 1 μg of lentiCRISPR/Cas9 MTDH sgRNA or LV-MTDH-ORF, 900 ng of PCMV delta R8.2 and 100 ng of pMD2.G. for 18 h in Opti-MEM medium (Gibco/Life Technologies). Fugene (Promega, Madison, WI, USA) was used as a transfection reagent according to the manufacturer’s instructions. After the transfection period, cells were refreshed with DMEM supplemented with 30% FBS to maintain the stability of the viral particles. The supernatant containing the particles were collected at 48 and 72 h post-transfection. Ultimately, the particles were concentrated with the Lenti-X^TM^ Concentrator (Takara, Kusatsu, Shiga, Japan) for 72 h at 4 °C, filtered, and stored at −80 °C.

Mammalian Cell Transduction: Parental cells SUM-149, SUM-190, and MCF-10A were transduced with the respective viral particles for each cell line. Briefly, cells (1 × 10^5^) were seeded in 6-well plates and allowed to grow until ~60% confluence. The transduction reaction was carried out with the viral particles of interest and 8 μg/mL Polybrene (Sigma-Aldrich) in Opti-MEM media (Gibco/Life Technologies) for 24 h. After transduction, the transduced cells were selected with 1 μg/mL of Puromycin (Sigma-Aldrich). 

### 4.3. RNA Isolation and Quantitative RT-qPCR Assays 

Total RNA from MTDH-edited and parental cell lines (SUM-149, SUM-190, and MCF-10A) and tumors from xenografted mice were extracted using the RNeasy Plus mini kit (Qiagen, Germantown, MD, USA) according to the manufacturer’s instructions. DNase digestion was performed after RNA isolation with DNase I according to the manufacturer’s protocol (Millipore Sigma, St. Louis, MO, USA). The samples were quantified with Nanodrop 1000 (Thermo Fisher Scientific, Waltham, MA, USA). cDNA was synthesized (1 µg RNA) using the iScript cDNA synthesis kit (Bio-Rad, Hercules, CA, USA). All reactions for cDNA synthesis were performed with the presence and absence of reverse transcriptase to exclude any possibility of DNA contamination. A quality control PCR with B2M primers was performed on an MJ Mini thermocycler (Bio-Rad) as follows: 12.5 µL 2× MM Jumpstart Taq (Millipore Sigma) or 2× MM Eco-Taq (Midsci, St. Louis, MO, USA), 1 µL 5 µM B2M of forward and reverse primers (Table 1), 2 µL 50 ng of cDNA or RT-minus reaction. The thermocycler program for the Jumpstart MM was utilized as follows: 94 °C for 2 min 1×, 94 °C for 30 s, 59 °C for 30 s, and 72 °C for 2 min 30× and a final extension of 72 °C for 5 min. The thermocycler program for the Eco-Taq MM was done as follows: 95 °C for 10 min 1×, 94 °C for 1 min, 59 °C for 30 s and 72 °C for 45 s 30×, and a final extension of 72 °C for 10 min. The PCR reactions were electrophoresed in a 2% agarose gel stained with SmartGlow safe green stain (Midsci) and photodocumented with the BioSpectrum Imaging System (UVP LLC, Upland, CA, USA). Additionally, amplification efficiency curves were performed for all primers. 

Those samples with a strong amplification band on RT-plus with its matched RT-minus reaction with no amplification band were selected to perform the qPCR. Quantitative PCR reactions were performed as follows 12.5 µL SSO SYBR Green Supermix (Bio-Rad), 300 nM of each primer, and 100 ng of cDNA in a 25 µL reaction. qPCR was performed on a CFX96 (Bio-Rad), and the thermocycler program was utilized as follows: an initial 95 °C for 3 min 1X, then forty cycles were run at 95 °C for 10 s, 60 °C for 30 s. The changes in gene expression were calculated using the 2^−ΔΔCt^ method as described in [65,67,68] using triplicate cDNA samples from two individual experiments. All primers were designed using the following websites https://www.idtdna.com/pages, https://primer3.ut.ee, http://biotools.nubic.northwestern.edu/OligoCalc.html, and https://blast.ncbi.nlm.nih.gov/Blast.cgi (all accessed on 3 June 2022) [69,70,71,72]. Primers were synthesized by Millipore Sigma, and their information can be found in Table 1.

### 4.4. Immunoblotting for Cells and Tumors

Cells and flash-frozen primary tumors from xenografted mice were lysed on ice and proteins were extracted using a lysis buffer containing 50 mM HEPES pH 7.0, 250 mM NaCl, 2 mM EDTA, 0.5% Igepal, 2 mM Na_3_VO_4_, 25 mM β-glycerol phosphate, 50 nM NaF, and cOmplete^TM^ Mini Protease Inhibitor Cocktail (Sigma-Aldrich). Total protein was quantified using the Precision Red protein assay kit (Cytoskeleton, Inc. Denver, CO, USA). Equal total protein amounts (10 μg) were subjected to separation by SDS-PAGE gels and transferred onto a PVDF membrane. After blocking with 5% milk, the membrane was incubated with the indicated primary antibodies in 5% bovine serum albumin (BSA) at 4 °C overnight at a dilution of 1:1000. The primary antibodies used were Anti-LYRIC/AEG1 (#ab124789, Abcam, Cambridge, MA, USA), AKT (#9272, CST, Danvers, MA, USA), p-AKT (Ser473) (#4060, CST), p44/42 MAPK (ERK1/2) (#9102, CST), phospho-p44/42 MAPK (p-ERK1/2 Thr202/Tyr204) (#4370, CST), STAT3 (#4904, CST), p-STAT3 (Tyr705) (#9145, CST), p-STAT3 (Ser727) (#9134, CST), JAK2 (#3230, CST), phospho-JAK2 (Tyr1007/1008) (#3771, CST), β-tubulin (#86298, CST) and β-actin (#A1978, Sigma-Aldrich). The membranes were then incubated with a secondary antibody according to their respective antibody species at a dilution of 1:10,000 or 1:20,000 for 1 h at room temperature (RT). The membrane was then developed with the Pierce TM ECL Western Blot Substrate kit (Thermo Fisher, Waltham, MA, USA) and visualized using the BioSpectrum Imaging System (UVP LLC). The integrated density of the bands of interest was quantified using ImageJ software (NIH, Bethesda, Maryland). Quantification of each protein was ensured by normalizing the integrated densities of bands of interest for all antibodies to the integrated density of the same immunoblotted lysate for β-acting or β-tubulin as described by us [40,65,73,74]. Arbitrary units are equal to the normalized integrated density of each protein relative to the control or parental cell line.

### 4.5. Proliferation Assay

Parental or MTDH-edited cells (SUM-149, SUM-190 and MCF-10A) were seeded in a 96-well plate with a density of 2.0 × 10^3^ cells/well for 24, 48 and 72 h. Proliferation was evaluated using the CyQUANT^®^ NF Cell Proliferation Assay Kit (Invitrogen, Waltham, MA, USA). Fluorescence was measured using a GloMax^®^ Explorer microplate reader (Promega) in the 500–550 nm range. The experiments were carried out in triplicate at least three times.

### 4.6. Colony Formation Assay

SUM-149 MTDH edited and parental cells were seeded in triplicate at 200 cells/well in a 24-well plate containing Ham’s F-12 with 10% FBS and incubated for 10 days. For SUM-190, MTDH edited and parental cells were seeded at 1 × 10^3^ cells/well in triplicate in a 6-well plate containing Ham’s F-12 with 2% FBS and incubated for 14 days changing the media every 4–5 days as described by [75]. After the incubation period at 37 °C, cells were fixed with methanol, washed with 1X phosphate saline buffer (PBS), stained with crystal violet for 5 min at RT, washed with water, and left to dry overnight. Colonies containing >50 cells were counted and analyzed using the Cytation 10 Confocal Imaging Reader (Agilent/BioTek, Santa Clara, CA, USA).

### 4.7. Wound Healing Assay

The cell capacity to migrate was evaluated by seeding parental or MTDH-edited cells (SUM-149, SUM-190, and MCF-10A) in two-well silicone inserts with a defined cell-free gap wound plate (Ibidi USA Inc., Madison, WI, USA). For SUM-149, 4 × 10^4^ cells/well were seeded, 8 × 10^4^ cells/well for SUM-190, and 1 × 10^5^ cells/well for MCF-10A. All cell lines were cultured in their respective complete culture medium for 24 h. After the incubation period, the insert was removed and the cells were allowed to migrate for 24 h at 37 °C as described by us in [74]. For MCF-10A cells, a complete culture medium with 2% HS was used to perform the assay. The cells were then fixed with 4% paraformaldehyde for 15 min, washed with 1X PBS, permeabilized with 0.1% Triton X-100 for 15 min at RT, and blocked with 1% BSA. Cells were stained for 1 h with a 1X rhodamine-phalloidin solution (Invitrogen^TM^/Life Technologies) to visualize actin filaments (F-actin). After washing three times with 1X PBS, cells were incubated with 1 µg/mL of DAPI (Life Technologies) for nuclear staining. Cell migration was quantified by measuring the distance (µm) between the wound edges using Olympus CellSense Imaging Software (Center Valley, PA, USA) on micrographs at a magnification of 4×. Fluorescence images were obtained at a magnification of 20× using the Cytation 10 Confocal Imaging Reader (Agilent/BioTek). 

### 4.8. Invasion Assay

Cell invasion was measured using the BD BioCoat Matrigel^TM^ invasion assay (BD Biosciences, San José, CA, USA). SUM-149 wild-type (WT) and MTDH-edited quiescent cells (1 × 10^5^) were seeded in the upper chambers and incubated at 37 °C for 24 h to allow invasion into 10% FBS medium (chemoattractant). After the incubation period, cells in the top chambers were removed with a cotton swab and cells attached to the bottom surface of the membrane were fixed and stained with propidium iodine (Sigma-Aldrich) as previously described in [65,74]. Cells were quantified with Gen5 Data Analysis Software (Agilent/BioTek) with a montage of micrographs obtained using the Cytation 10 Confocal Imaging Reader (Agilent/BioTek) at a magnification of 20×. 

### 4.9. IBC-Dderived Tumor Spheroids Assay

SUM-149 and SUM-190 (WT and MTDH-edited) cells were seeded in triplicate in 6-well ultralow attachment plastic plates (Corning^®^). For SUM-149, a density of 4 × 10^4^ cells/well was used and 8 × 10^4^ cells/well for SUM-190. All cell lines were cultured in their respective complete media supplemented with 2.25% polyethylene glycol (PEG-800) (Sigma-Aldrich) as described by [34,76]. Cells were incubated at 37 °C for a period of 96 h. After the incubation period, the micrographs were captured using the Cytation 10 Confocal Imaging Reader (Agilent/BioTek). Gen5 Data Analysis Software (Agilent/BioTek) was used to calculate the quantity and area of tumor spheroids.

### 4.10. In Vivo Study

The study was approved by the Universidad Central del Caribe (UCC) Institutional Animal Care and Use Committee (IACUC) (Animal Welfare Assurance #D16-00343) and was carried out following IACUC guidelines.

Female severe combined immunodeficient mice (SCID) (Charles River Laboratories International, Wilmington, MA, USA) between 21 and 28 days of age were housed under specific pathogen-free conditions. Mice received an irradiated AIN 76-A phytoestrogen-free diet (Tek Global, Harlan Teklad, Madison, WI, USA) and water ad libitum. To test the effects of MTDH silencing in IBC tumor formation and progression, we injected 1.5 × 10^6^ WT SUM-149 and MTDH-edited cells into the lower right mammary fat pad of female mice in a 100 µL volume dilution (1:1) of reduced growth factor Matrigel (BD Biosciences) and serum-free media, as previously described in [67,73]. Group allocation was made randomly: (a) mice injected with WT cells (*n* = 10) and (b) mice injected with MTDH-edited cells (*n* = 10). One week after injection, the weight and tumor volume of the mice were measured weekly for 10 weeks. Tumor volume (mm^3^) was measured with a caliper and calculated as follows: [(width)^2^ × length)/2] as described in [77]. At the end of the study, the tumors and both lungs were excised and kept in optimal conditions for future experiments. 

### 4.11. Lung Sectioning and Hematoxylin & Eosin (H & E) Staining

Sectioning: The excised lungs were left in 10% neutral-buffered formalin (NBF) overnight and then immersed in a solution of 20% sucrose in 1X PBS. The lungs were then placed in optimal cutting temperature compound (OCT) (TFM^TM^, General Data Company Inc., Cincinnati, OH, USA). Sectioning was performed using a Leica CM 1860 cryostat (Leica Byosystems, Deer Park, IL, USA) by cutting 8 μm sections of the left and right lung of mice injected with WT SUM-149 or MTDH-edited cells (*n* = 7/group). The sections were mounted on slides with a subbing solution as described in [78]. 

Hematoxylin & Eosin (H & E) staining: First, tissues were stained with Mayer hematoxylin solution (Sigma-Aldrich) for 10 min at RT and then rinsed with running tap water for 10 min until the water was colorless. The tissues were then rinsed in 95% alcohol for 10 s. Then, the staining was performed with Eosin Y with Phloxine B (Sigma-Aldrich) solution for 30 s and rinsed in ascending series of ethanol (70%, 95%, and 100%). Finally, tissues were processed in xylene 1 min twice and mounted using VectaMount^®^ permanent mounting medium (Vector Labortories, Newark, CA, USA). A pathologist analyzed H&E-stained slides and the micrographs were captured at a magnification of 20×. 

### 4.12. Statistical Analysis 

The *p*-value for the in vitro studies was calculated using Student’s *t*-tests, ordinary-one way or two-way analysis of variance (ANOVA) with the Bonferroni multiple comparison test. Gene expression studies for each cell line or tumor were evaluated using the 2^−ΔΔCt^ formula by comparing their relative gene expression to the reference genes. Data are expressed as Mean ± S.E.M. Each experiment was carried out in three or more independent biological replicates. Statistical analyses were performed using Graph Pad Prism v.9.0 (San Diego, CA, USA), and differences were considered significant when *p* < 0.05.

In vivo studies: Summary statistics were performed to describe the tumor volume and weight. The normality evaluation was performed using the Shapiro–Wilk test. The presence of outliers was verified using a generalized extreme studentized deviation (ESD) test. Mean changes were evaluated over time using a general linear model with repeated measures. Model diagnostics were performed as well, i.e., Cooks distances and DF betas were ascertained to access influential values. A pairwise comparison was performed to determine the difference between the groups and each week. Only if the data did not follow a normal distribution, Wilcoxon–Mann–Whitney tests were performed. The significance level (α) was set to 0.05. Analyses were performed using the statistical software R version 4.1.0.

## 5. Conclusions

Our findings identify for the first time a potential role of MTDH in promoting IBC proliferation, migration, invasion, and metastasis by regulating NF-κB and STAT3. Furthermore, we showed that MTDH plays a role in the invasive capacity of IBC promoting metastasis to the lung. Altogether, these findings open the possibility of a new target to better understand the molecular biology of IBC. MTDH has the potential to serve as a diagnostic target to diagnose and treat the progression of this deadly disease in IBC patients.

## Figures and Tables

**Figure 1 ijms-24-04694-f001:**
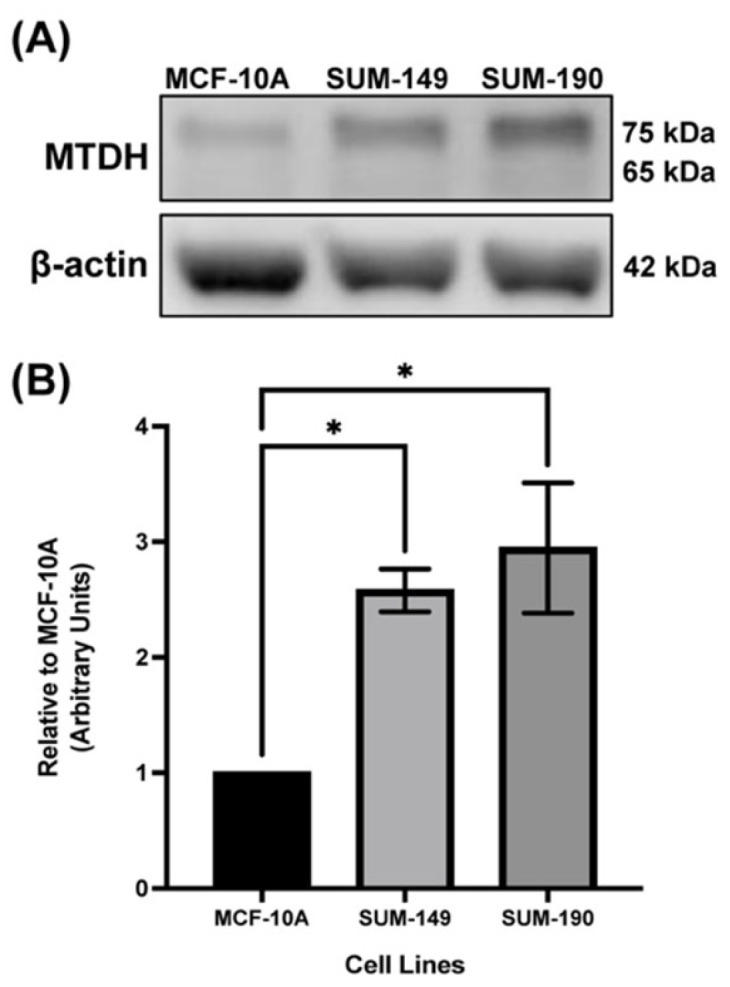
Basal MTDH expression in IBC cell line models and non-cancerous mammary epithelial cells. (**A**) Total protein lysates of MCF-10A, SUM-149, and SUM-190 were probed using the indicated antibodies. β-actin was used as a loading control. (**B**) Densitometry quantification of MTDH in MCF-10A, SUM-149, and SUM-190 cells. Data are expressed relative to MCF-10A cells. Ordinary one-way ANOVA with Bonferroni’s multiple comparison test, * *p* < 0.05. Results are shown as Mean ± SEM. The experiment was performed at least three times. The approximate molecular weight of MTDH bands are stated to be 75 kDa and 65 kDa, according to the manufacturer’s description.

**Figure 2 ijms-24-04694-f002:**
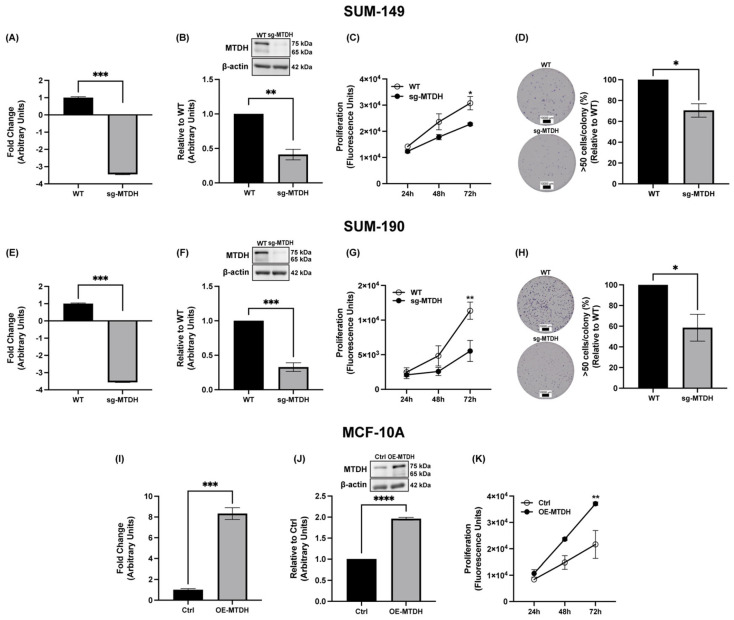
The proliferation and colony formation capacity of IBC cells was suppressed after MTDH knockout. (**A**,**B**) MTDH qPCR, immunoblot, and densitometry analysis of MTDH in wild-type (WT) SUM-149 cells after knockout of the MTDH gene (sg-MTDH) using CRISPR/Cas9. (**C**) Proliferation of SUM-149 WT and sg-MTDH cells. (**D**) Clonogenic capacity of SUM-149 WT and sg-MTDH cells. (**E**,**F**) MTDH qPCR, immunoblot, and densitometry analysis of MTDH in SUM-190 WT and sg-MTDH cells. (**G**) Proliferation of SUM-190 WT and sg-MTDH cells. (**H**) Clonogenic capacity of SUM-190 WT and sg-MTDH cells. (**I**,**J**) MTDH qPCR, immunoblot, and densitometry analysis of MTDH in MCF-10A cells after transduction with a control (Ctrl) vector or MTDH cDNA (OE-MTDH). (**K**) Proliferation of MCF-10A Ctrl and OE-MTDH cells. MTDH qPCRs were performed using HPRT and *RPL13A* as reference genes, and changes in gene expression were calculated using the 2^−ΔΔCt^ method (*** *p* < 0.001). Immunoblots were performed with β-actin used as loading control and analyzed with Student’s *t*-test (** *p* < 0.01, *** *p* < 0.001 and **** *p* < 0.0001). Cell proliferation was tested for 24, 48, and 72 h and analyzed with two-way ANOVA with Bonferroni’s multiple comparison test; * *p* < 0.05, ** *p* < 0.01. SUM-149 and SUM-190, were incubated for 10 and 14 days, respectively. The colonies were stained with crystal violet, and the relative number of colonies with >50 cells were plotted. A montage of images from each well was obtained using the Cytation 10 Confocal Imaging Reader; scale bar = 10,000 µm. A Student’s *t*-test was performed; * *p* < 0.05. Results are shown as Mean ± SEM. The experiments were carried out at least three times.

**Figure 3 ijms-24-04694-f003:**
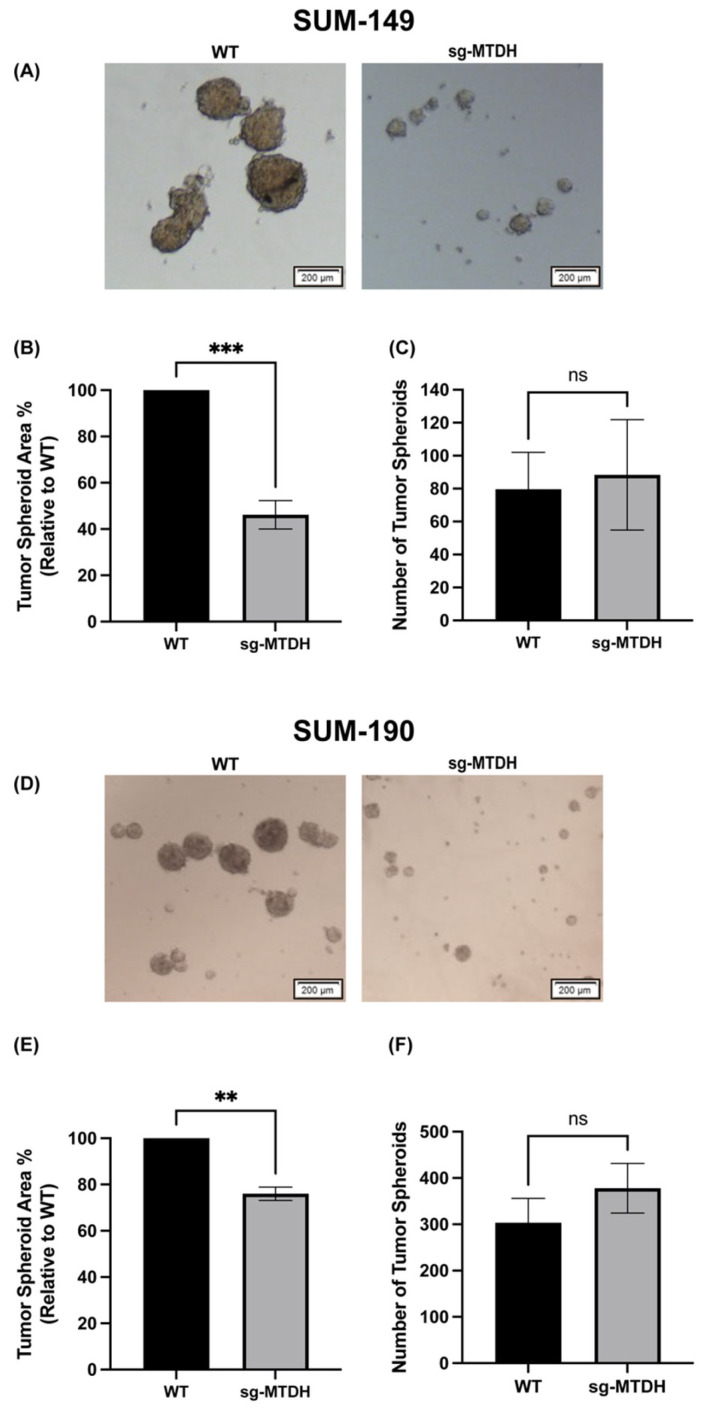
IBC-derived tumor spheroids size was significantly reduced in MDTH silenced IBC cells. SUM-149 and SUM-190, WT and sg-MTDH cells, were grown in ultralow attachment plates for 96 h to allow IBC-derived tumor spheroids to form. (**A**) Micrographs of SUM-149 WT and sg-MTDH represent a region of an image montage of each condition. (**B**) Area and (**C**) number of SUM-149 WT and sg-MTDH tumor spheroids. (**D**) Micrographs of SUM-190 WT and sg-MTDH represent a region of an image montage of each condition. (**E**) Area and (**F**) number of SUM-190 WT and sg-MTDH tumor spheroids. The area and number of tumor spheroids were quantified using Gen5 Data Analysis Software. Scale bar = 200 µm. Student’s *t*-test ** *p* < 0.01, *** *p* < 0.001, ns: not significant. Results are shown as Mean ± SEM. The experiment was carried out at least three times.

**Figure 4 ijms-24-04694-f004:**
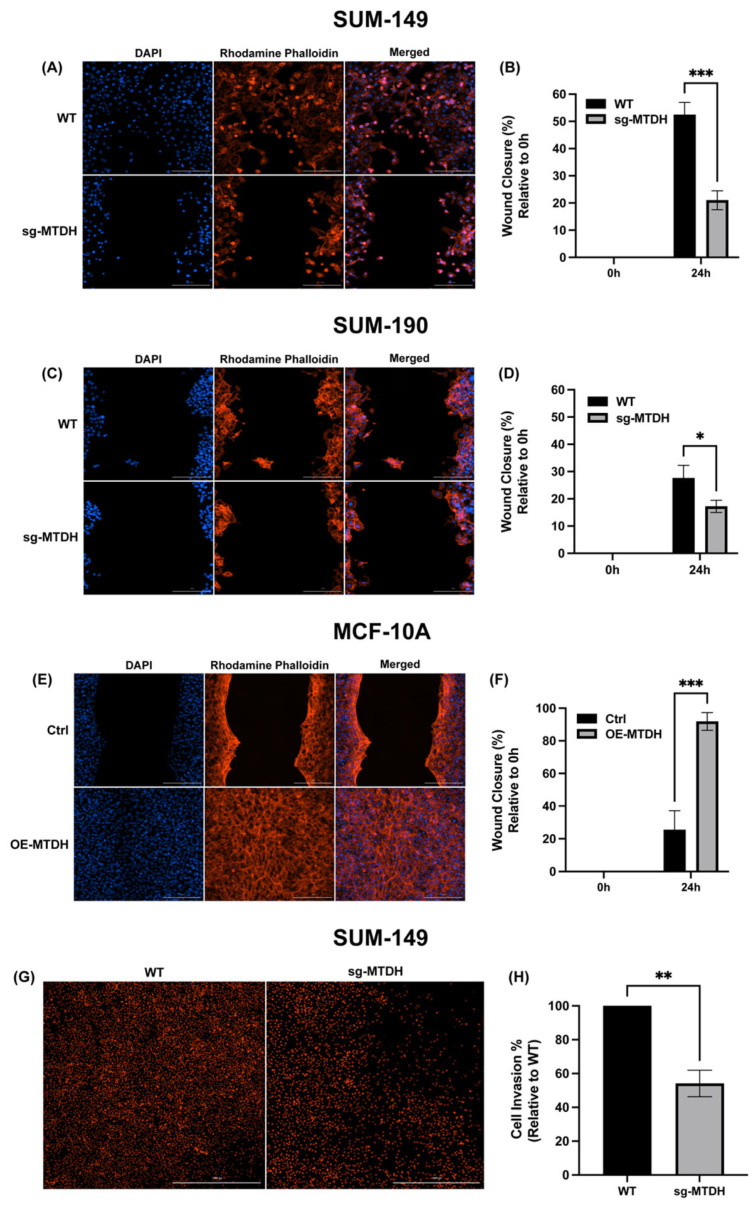
MTDH depletion modulates the motility and invasive capacity of IBC cells. (**A**,**B**) Micrographs of the wound healing assay and wound closure (%) results of SUM-149 WT and sg-MTDH with cells seeded in Ibidi^®^ silicone insert plates and incubated for 24 h. (**C**,**D**) Micrographs from the wound healing assay and wound closure results for SUM-190 WT and sg-MTDH. (**E**,**F**) Wound healing assay micrographs and wound closure (%) results of MCF-10A Ctrl and OE-MTDH cells. The width of the wound was calculated by measuring the distance between the edges of the wound on the micrographs taken at 4× using Olympus CellSense Imaging Software. The percentage of wound closure was calculated by normalizing the results of each cell line to 0 h and then normalizing the results of each edited cell line to its parent cell line. After incubation, the nuclei were stained with DAPI (blue), and the actin cytoskeleton was stained with rhodamine-phalloidin (red). Micrographs of stained cells were taken at a magnification of 20× using the Cytation 10 Confocal Imaging Reader; scale bar = 200 µm. Two-way ANOVA with Bonferroni’s multiple comparison test; * *p* < 0.05, *** *p* < 0.001. (**G**,**H**) SUM-149 WT and sg-MTDH cells were incubated in BD BioCoat Matrigel^TM^ invasion chambers for 24 h, and the invading cells were stained with PI. Micrographs represent an image montage of each condition taken at 20× with the Cytation 10 Confocal Imaging Reader; scale bar = 10,000 µm. Two-way ANOVA with Bonferroni’s multiple comparison test; * *p* < 0.05, *** *p* < 0.001. A Student’s *t*-test was performed; ** *p* < 0.01. Results are shown as Mean ± SEM. The experiments were carried out at least three times. Micrographs at 4× magnification are shown in Appendix A.

**Figure 5 ijms-24-04694-f005:**
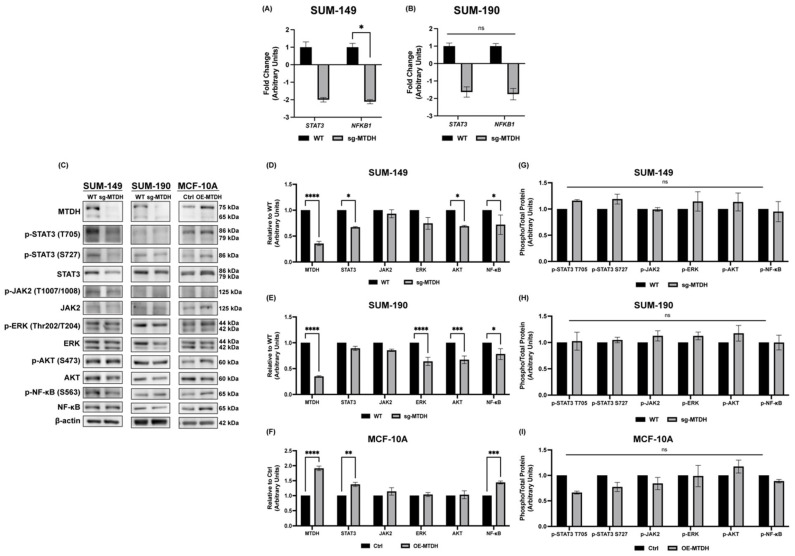
MTDH knockout decreases STAT3, NF-κB, and AKT expression. (**A**) *STAT3* and *NFKB1* qPCR for SUM-149 WT and sg-MTDH. (**B**) *STAT3* and *NFKB1* qPCR for SUM-190 WT and sg-MTDH. qPCRs were performed using *HPRT* and *RPL13A* as reference genes, and changes in gene expression were calculated using the 2^−ΔΔCt^ method (* *p* < 0.05, ns: not significant). (**C**) Immunoblots of whole cell lysates of parental and edited cells of SUM-149, SUM-190, and MCF-10A were performed for the indicated proteins. (**D**–**I**) Densitometry quantification of total and phospho-proteins in each parental (WT or Ctrl) and edited (sg-MTDH or OE-MTDH) cell lines (SUM-149, SUM-190, and MCF-10A). After the densitometry analysis, the intensities were normalized to their loading control and plotted relative to their parental cells. A two-way ANOVA with Bonferroni’s multiple comparison test was performed; * *p* < 0.05, ** *p* < 0.01, *** *p* < 0.001 and **** *p* < 0.0001, ns: not significant. Results are shown as Mean ± SEM. The experiments were carried out at least three times.

**Figure 6 ijms-24-04694-f006:**
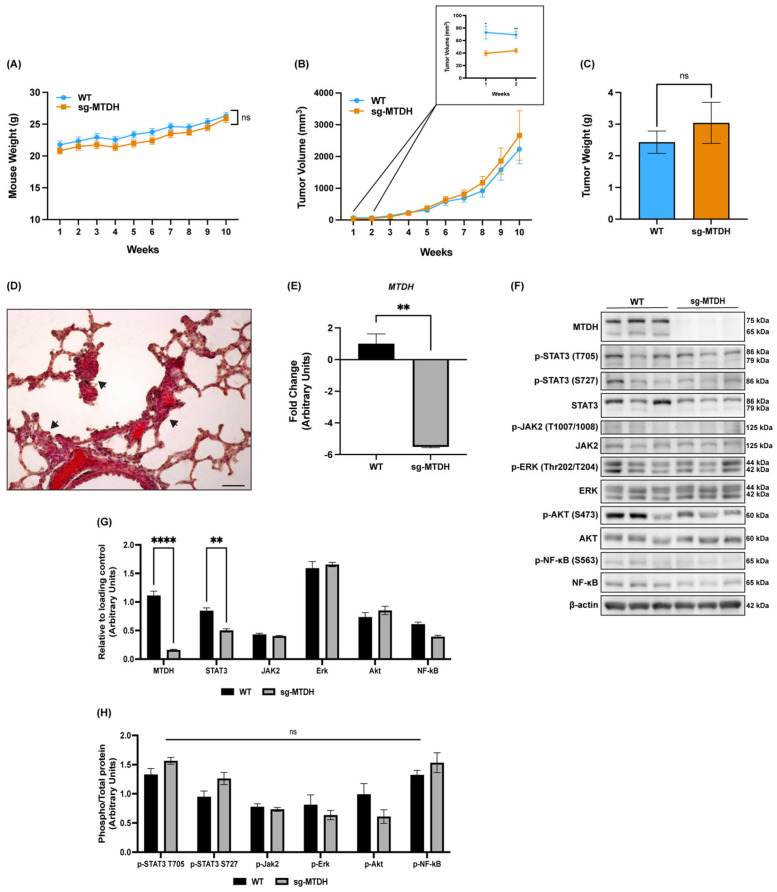
MTDH knockout delayed tumor development, reduced lung metastasis, and decreased STAT3 expression in a SUM-149 xenograft model. Xenograft IBC models were generated by injecting SUM-149 WT and SUM-149 sg-MTDH cells into SCID mice (*n* = 10/group). (**A**) The body weight of the mice and (**B**) the tumor volume (mm^3^) were monitored for 10 weeks after injection. A pairwise comparison was made to determine the difference between the groups and each week to compare the average tumor volume. Wilcoxon-Mann-Whitney tests were performed only if the data did not follow a normal distribution; ** *p* < 0.01, ns: not significant. (**C**) The tumor weight of the mice was measured at the end of the study, *t*-test; ns: not significant. (**D**) H & E staining of a lung tissue section of a tumor of the SUM-149 WT group (*n* = 7/group) at the end of the study shows small deposits of epithelial-like cells (arrows) in the alveolar parenchyma. Scale bar = 200 µm. (**E**) *MTDH* qPCR for tumors of SUM-149 WT and sg-MTDH mice. qPCR was performed using *HPRT* and *RPL13A* as reference genes, and changes in gene expression were calculated using the 2^−ΔΔCt^ method (** *p* < 0.01). (**F**) Immunoblots of SUM-149 WT and sg-MTDH tumors, each lane representing a different mouse (*n* = 3/group). Tumor lysates were probed for the indicated proteins. (**G**,**H**) Densitometry quantification analysis of total and phospho-proteins; the intensities were normalized to its loading control and plotted. A two-way ANOVA with Bonferroni’s multiple comparison test was performed; **** *p* < 0.0001, ns: not significant. Results are shown as Mean ± SEM.

**Table 1 ijms-24-04694-t001:** List of the genes and primers used in the RT-qPCR analysis.

Gene	Accession Number	Primer Sequence
**Beta-2-Microglobulin** ** *(B2M)* **	NM_004048.4	FW 5′TGCTGTCTCCATGTTTGATGTATCT3′ REV 5′TCTCTGCTCCCCACCTCTAAGT3′
**Hypoxanthine** **Phosphoribosyltransferase** ** *(HPRT)* **	NM_000194.3	FW 5′CCTGGCGTCGTGATTAGTGAT 3′ REV 5′AGACGTTCAGTCCTGTCCATAA 3′
**Metadherin** ** *(MTDH)* **	NM_178812.4	FW 5′GAAACTGTCCGAGAAGCCCA3′ REV 5′GTCAATCTCTGGTGGCTGCT3′
**Signal Transducer and** **Activator of Transcription 3** ** *(STAT3)* **	NM_139276.3	FW 5′GCCAATTGTGATGCTTCC 3′ REV 5′TGGGTCTCTAGGTCAATCT 3′
**Ribosomal Protein L13a** ** *(RPL13A)* **	NM_012423.4	FW 5′TGAAGCCTACAAGAAAGTTTGCCT 3′ REV 5′TAGCCTCATGAGCTGTTTCTTCTT 3′
**Nuclear factor kappa** **B subunit 1** ** *(NFKB1)* **	NM_003998	FW 5′ATAGCCTGCCATGTTTGCTGCT 3′
		REV 5′TGCCAATGAGATGTTGTCGTGC 3′

## Data Availability

Not applicable.

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
