# Peer review of "Metadherin Regulates Inflammatory Breast Cancer Invasion and Metastasis"

_ijms, 2023, doi:10.3390/ijms24054694_

Round 1

Reviewer 1 Report

The manuscript entitles “ Metadherin regulates Inflammatory Breast Cancer Invasion and Metastasis ” by Gabriela Ortiz-Soto et al, identified a functional role for MTDH in the aggressiveness of IBC by affecting cell proliferation, tumor spheroid formation, migration, and invasion. The data shown are of potential interest. However, the authors have to address and consider several points to publish this manuscript as follows:

1. The authors examined biomarkers in several classical pathways that are associated with promoting tumorigenesis and cell proliferation, what was the purpose of designing the experiment in this way? As MTDH has been shown to be closely associated with multiple signaling pathways, the results by immunoblots alone can only show that MTDH is associated with STAT3 and NF-κB and cannot confirm “a crosstalk between STAT3 and NF-κB signaling pathways”. The authors can try to focus the study on the most important pathway might lead to better conclusions.

2. In Vivo Studythe absence of MTDH dose not reduce tumor volume or tumor weight. This is contrary to the results of the in vitro results, especially the results of 3D culture. Please provide a more detailed explanation in the “Discussion”.

3. The original images of western blot don’t show the marker bands. The results are not rigorous enough.

4. The Reference 37 need to be standardized.

Author Response

  1. The authors examined biomarkers in several classical pathways that are associated with promoting tumorigenesis and cell proliferation, what was the purpose of designing the experiment in this way? As MTDH has been shown to be closely associated with multiple signaling pathways, the results by immunoblots alone can only show that MTDH is associated with STAT3 and NF-κBand cannot confirm “a crosstalk between STAT3 and NF-κB signaling pathways”. The authors can try to focus the study on the most important pathway might lead to better conclusions.

  • In a previous proteomics study, our group identified and validated nine overexpressed cell surface biomarkers in IBC that have been correlated with cancer invasion (PMID: 35681787). MTDH was identified as one of these biomarkers that correlated with cancer progression and metastasis. Since there were no previous studies on the role of MTDH in IBC, we sought to study and characterize the functionality of this protein in the invasiveness of IBC models. Please refer to lines 51-54 in the Introduction and lines 88-92 in the Results section.
  • We have included additional references of previous studies that describe the direct interactions between NF-κB and STAT3 (Please refer to lines 430-432). Also, a statement was added to highlight the importance of understanding the regulation of NF-κB and STAT3 in IBC in relation to MTDH (Please refer to lines 258-262).

  1. In Vivo Study, the absence of MTDH does not reduce tumor volume or tumor weight. This is contrary to the results of the in vitro results, especially the results of 3D culture. Please provide a more detailed explanation in the “Discussion”.

  • We have included a statement explaining a potential explanation of the discrepancies of the in vitro and in vivo results (Please refer to lines 442-447). Even though, there were no significant differences in the tumor volume or weight we showed the effects of MTDH in reduction of IBC cell invasion and metastasis (Please refer to lines 447-483).

  1. The original images of western blot don’t show the marker bands.The results are not rigorous enough.

  • In our laboratory, the protocol for Western Blot allows us to cut the membranes to identify various proteins from a single blot. In the original Western Blot image file, some of the membranes show the molecular marker, for example, in the blots for Figure 1 (MTDH & β-actin) and Figure 5C (p-STAT3 T705). Additionally, in each Western Blot figure we included the molecular weights for each protein to facilitate the reader in the identification of the proteins.

  1. The Reference 37 need to be standardized.

  • Reference 37 was standardized, and it was changed to Reference 40. Please refer to line 844.

Author Response

  1. Abstract: NF-κB and STAT3 results is missing in abstract as it is one of interesting finding of this study.

  • We have included NF-κB and STAT3 as findings in the abstract. Please refer to line 29.

  1. Figure 1: Arbitrary units is not clear. It is better to write about this either in legend or fold change wrt to …

  • A statement defining the arbitrary units has been added to the Methods section. Please refer to lines 594-595.

  1. Figure 1: In gel, about 75kda and 65kda should be mentioned in this figure legends

  • A statement including 75kDa and 65kDa has been added to the figure 1 legend. Please refer to lines 108-109.

  1. Results 2.5: STAT3 or NF-κB1 gene expression was not significantly change in SUM-190 while reduced in SUM-149 model. How authors justify these two contrasting results as both are IBC models.

  • A statement has been added to the discussion addressing the contrasting results between the gene expression of both IBC models. Please refer to lines 404-407.

  1. Densitometry analysis showed the reduction of NF-κB protein expression in SUM-190 but gene expression was not significant change in this model. These results need to be discussed with justification in discussion section with previous publication in this directions.

  • Please refer to lines 408-413.

Reviewer 3 Report

see the report

Author Response

  1. as explained in lines 77-79, it is better to write in line 27,” Our results demonstrate that the absence of MTDH”.

  • The suggested phrasing was corrected. Please refer to lines 27-28.

Round 2

Reviewer 1 Report

The queries addressed by the authors adequately.

Author Response

Thank you